# APF-DPPO: An Automatic Driving Policy Learning Method Based on the Artificial Potential Field Method to Optimize the Reward Function

**Junqiang Lin [1], Po Zhang [1], Chengen Li [1], Yipeng Zhou [3], Hongjun Wang [1,2,*] and Xiangjun Zou [1,2,4]**

1   College of Engineering, South China Agricultural University, Guangzhou 510642, China;
    jqlin@stu.scau.edu.cn (J.L.); zhangtupi@foxmail.com (P.Z.); lichengen4549@gmail.com (C.L.);
    xjzou1@163.com (X.Z.)
2   Guangdong Laboratory for Lingnan Modern Agriculture, Guangzhou 510642, China
3   Maritime Academy, Ningbo University, Ningbo 315000, China; zhouyipeng1717@163.com
4   Foshan-Zhongke Innovation Research Institute of Intelligent Agriculture and Robotics, Foshan 528000, China
*   Correspondence: xtwhj@scau.edu.cn; Tel.: +86-186-6568-6187

**Abstract:** To address the difficulty of obtaining the optimal driving strategy under the condition of a complex environment and changeable tasks of vehicle autonomous driving, this paper proposes an end-to-end autonomous driving strategy learning method based on deep reinforcement learning. The ideas of target attraction and obstacle rejection of the artificial potential field method are introduced into the distributed proximal policy optimization algorithm, and the APF-DPPO learning model is established. To solve the range repulsion problem of the artificial potential field method, which affects the optimal driving strategy, this paper proposes a directional penalty function method that combines collision penalty and yaw penalty to convert the range penalty of obstacles into a single directional penalty, and establishes the vehicle motion collision model. Finally, the APF-DPPO learning model is selected to train the driving strategy for the virtual vehicle, and the transfer learning method is selected to verify the comparison experiment. The simulation results show that the completion rate of the virtual vehicle in the obstacle environment that generates penalty feedback is as high as 96.3%, which is 3.8% higher than the completion rate in the environment that does not generate penalty feedback. Under different reward functions, the method in this paper obtains the highest cumulative reward value within 500 s, which improves 69 points compared with the reward function method based on the artificial potential field method, and has higher adaptability and robustness in different environments. The experimental results show that this method can effectively improve the efficiency of autonomous driving strategy learning and control the virtual vehicle for autonomous driving behavior decisions, and provide reliable theoretical and technical support for real vehicles in autonomous driving decision-making.

**Keywords:** deep reinforcement learning; proximal policy optimization; autonomous driving; driving strategy; artificial potential field method; reward function; transfer learning

## 1. Introduction

Autonomous driving technology is a high-tech technology that is based on sensors, the Internet of Things, and artificial intelligence [1]. This technology has important application value in building smart transportation, such as alleviating traffic pressure, reducing traffic accidents, and saving energy consumption [2–5]. With gradual improvement in the informatization degree of society, people have achieved increasingly higher functional requirements for cars. Most traditional driving strategies adopt manual design and mathematical modeling methods [6], which cannot meet the needs of intelligent vehicle driving technology. An intelligent driving system with autonomous learning and autonomous decision-making can effectively compensate for the defects of traditional control methods.

Presently, research on autonomous driving technology is divided into four parts: perception and positioning, behavioral decision-making, path planning, and control execution [7]. In a complex and changeable driving environment, ensuring that the vehicle can quickly make driving decisions under the premise of safety is a key problem to be urgently solved with current automatic driving technology; it is also the core embodiment of intelligent transportation. Therefore, it is very important to improve the decision-making ability of vehicle autonomous driving.

In terms of vehicle autonomous driving decision-making, many scholars worldwide have carried out related research, focusing on three parts: task-driven rule limitation methods [8–11], data-driven deep learning methods [12,13], and experience-driven reinforcement methods [14–17]. The task-driven rule definition method is to artificially construct a driving behavior rule library according to traffic laws and then to select driving behaviors according to environmental information, which has good interpretability and stability. The Talos self-driving car developed by MIT [18] and the junior self-driving car jointly developed by Volkswagen and Stanford University [19] use this method to conduct self-driving experimental research on vehicles. However, this method has poor generalization performance and a weak understanding of the driving environment. When road conditions change, the rule base needs to be redesigned, which cannot meet the time-varying and diverse needs of autonomous driving. Deep learning (DL) methods have strong environmental perception and data analysis capabilities and can automatically obtain low-dimensional representations of high-dimensional data through algorithms, thereby obtaining a large amount of useful knowledge that is suitable for complex driving environments [20]. Chen et al. [21] utilized deep convolutional neural networks for autonomous driving decision learning training for vehicles in virtual environments. Bojarski et al. [22] collected driving environment data through cameras and applied a convolutional neural network approach to realize end-to-end autonomous driving in the real world for the first time. However, this method needs to rely on a large amount of prior knowledge and lacks the ability of autonomous learning and autonomous decision-making, so it is difficult to adapt to the unknown and changeable driving environment. Reinforcement learning (RL) methods do not need to rely on environmental models and prior knowledge and can autonomously perceive the environment and continuously optimize their own behavior strategies in combination with environmental feedback [23]. However, traditional reinforcement learning methods are easily limited by the dimensions of state and action space and cannot adapt to high-dimensional, continuous-action driving environments. The rise of deep reinforcement learning provides a new idea for solving this problem, endows the vehicle with perception and decision-making capabilities, and greatly improves the efficiency of autonomous vehicle decision-making [24,25]. Xia et al. [26] improved the stability of the autonomous driving strategy model by 32% by improving the Q-learning algorithm and adding driving teaching data. Chae et al. [27] selected the DQN algorithm to train vehicles for autonomous driving decision-making and achieved good results. Jaritz et al. [28] employed the asynchronous superiority critic algorithm to obtain images of different road conditions through cameras as input signals to complete automatic driving decision control. The above method mainly adopts the discrete action space encoding method for the output action; the vehicle driving environment is relatively fixed; the driving strategy is relatively stable; and the driving action is relatively standardized. In the real driving process, the steering wheel angle, accelerator pedal strength, and brake pedal opening of the vehicle are encoded in a continuous action space, and the driving road conditions and obstacles are random and uncertain, so it is necessary to further improve the autonomous decision-making of the vehicle's capabilities and optimized autonomous driving strategies.

Based on the above problems, this paper selects vehicle autonomous driving as the research object and builds a virtual vehicle autonomous driving strategy learning system that is based on deep reinforcement learning (DRL). First, according to the physical structure of the vehicle, the random driving motion strategy of the virtual vehicle is set, and the state space is reasonably set by analyzing the actual vehicle driving behavior and environmental

information. The virtual vehicle action space (steering control and speed control) is set according to the dynamic and kinematic model of the vehicle. Second, a continuous reward function learning model is designed based on the idea of an artificial potential field (APF) and is integrated into the distributed proximal policy optimization (DPPO) framework, which is referred to as the APF-DPPO algorithm. To solve the range repulsion problem of the APF method, which affects the optimal driving strategy of the vehicle, a directional penalty function method that combines collision penalty and yaw penalty is proposed. By establishing a motion collision model of virtual vehicles, the results of motion collisions can be analyzed, direction penalties can be selectively given, and the decision-making ability of the vehicle's automatic driving can be continuously improved. Last, based on the Unity virtual simulation platform, the Machine Learning Agents (ML-Agents) plug-in is employed to establish interactive communication between the simulation environment and the Python API, and the APF-DPPO algorithm is utilized to learn and train the automatic driving strategy of the virtual vehicle, to continuously improve the automatic driving decision-making ability and to complete the task of autonomous vehicle driving. To further verify the effectiveness and rationality of the method in this paper, the transfer learning (TL) method is applied to transfer the resulting model to the comparative experiment for automatic driving, decision inference learning, and analysis of the experimental results. The main contributions and innovations of this paper are reflected in the following aspects:

1.  This work proposes a deep reinforcement learning-based self-driving policy learning method for improving the efficiency of vehicle self-driving decisions, which addresses the problem of difficulty in obtaining optimal driving policies for vehicles in complex and variable environments.
2.  The ideas of target attraction and obstacle rejection of the artificial potential field method are introduced into the distributed proximal policy optimization algorithm, and the APF-DPPO learning model is established to evaluate the driving behavior of the vehicle, which is beneficial to optimize the autonomous driving policy.
3.  In this paper, we propose a directional penalty function method combining collision penalty and yaw penalty, which transforms the range penalty of obstacles into a single direction penalty and selectively gives directional penalty by building a motion collision model of the vehicle, and this method effectively improves the efficiency of vehicle decision-making. Finally, the vehicles are trained to learn driving strategies and the resulting models are experimentally validated using TL methods.

The remainder of this paper is structured as follows: Section 2 introduces the overall framework of the system, the modeling of autonomous driving strategy optimization problems, and the theoretical background of research methods. Section 3 builds a vehicle autonomous driving strategy learning system and establishes an APF-DPPO-based learning system. The continuous reward function learning model describes the training process of the APF-DPPO algorithm. Section 4 designs the experimental scheme and analyzes the training result data in detail. Combined with the verification and analysis of the APF-DPPO algorithm model of TL, the decision control effect of the actual vehicle driving process under different conditions is introduced. Section 5 concludes the paper.

## 2. Research Methods

### 2.1. Overall Framework of the System

In this paper, the framework idea of reinforcement learning is applied to build a virtual vehicle autonomous driving strategy training learning and reasoning learning system. The overall architecture is shown in Figure 1. First, the vehicle simulation model and driving environment are built, and the vehicle state information and pictures of the learning environment are collected in real time through virtual ray sensors and cameras for feature extraction and then passed to the deep neural network for comparative analysis. The network parameters of the driving strategy are continuously updated. According to the physical structure of the vehicle, the driving action and motion range of the virtual vehicle are set. Second, based on the idea of the APF method, a continuous reward function learning

model is established; the design destination guidance function, the obstacle avoidance penalty function, and the time penalty function are constructed; and the driving strategy of the virtual vehicle is evaluated. Last, based on the Unity ML-Agents reinforcement learning simulation environment, the APF-DPPO algorithm is selected to learn and train the autonomous driving strategy of the virtual vehicle; the convolutional neural network is employed to iteratively update the network parameters of the virtual vehicle; and the control effect is fed back to the simulation system in real time for visual display. After completing the training and learning, the system will automatically generate a network model based on TensorFlow, which can quickly guide the virtual vehicle to safely reach the destination and complete the automatic driving task. The TL method is used to transfer the training result model to the comparison experiment under different conditions for vehicle automatic driving decision inference learning. In addition, to further improve the flexibility and operability of the system, an interactive control interface is built. Users can select a specific environment to simulate intelligent vehicles for autonomous driving behavior decision-making and to obtain relevant driving information data to provide a reliable basis for actual vehicles to make autonomous driving decisions.

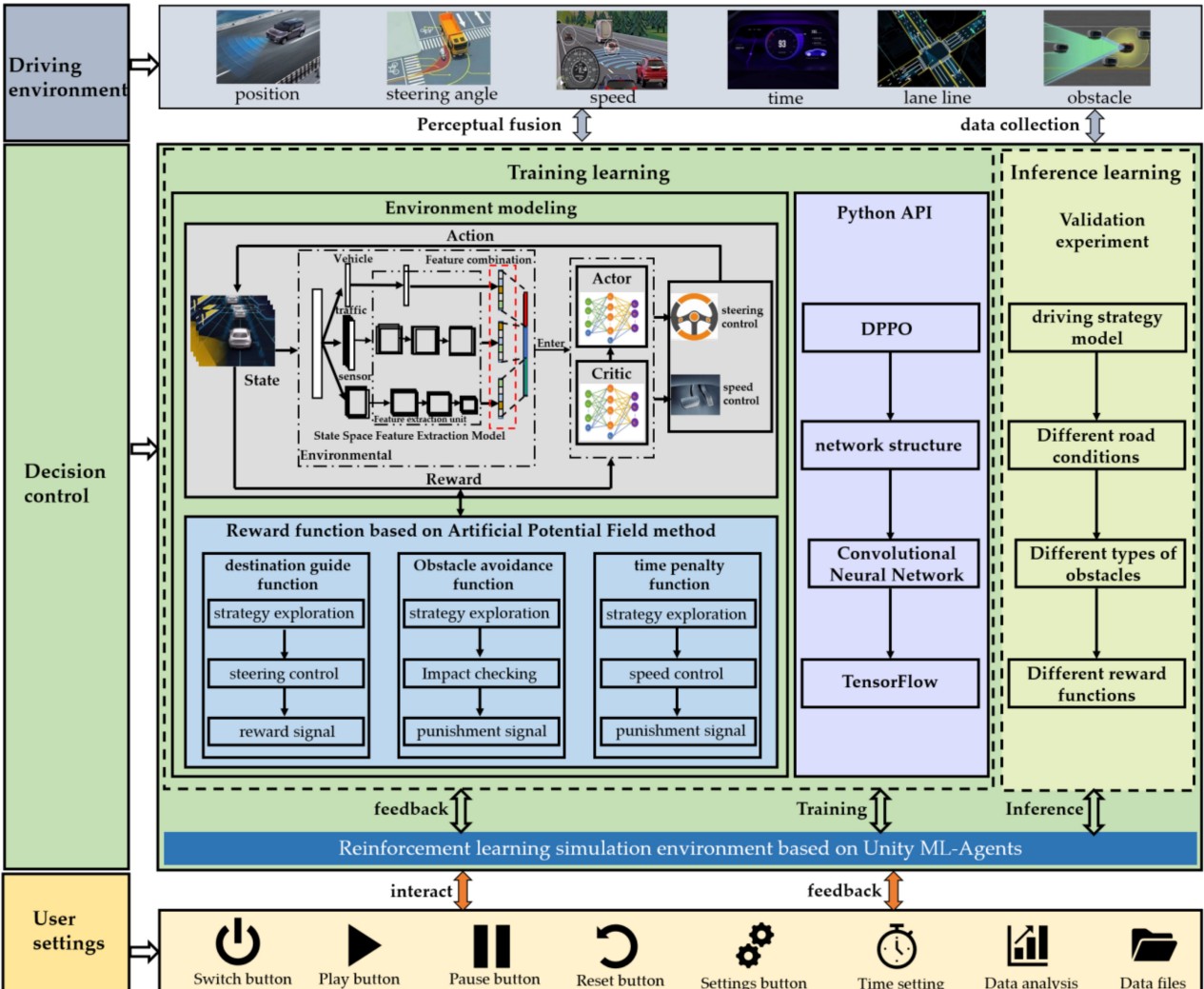

**Figure 1.** Overall architecture diagram.

## 2.2. Modeling of the Automatic Driving Strategy Optimization Problem

Due to the large number of unpredictable driving scenarios in the real world, an end-to-end decision control method can effectively solve complex and changeable environ-

mental problems. Deep reinforcement learning (DRL) is an end-to-end learning method that combines the perception ability of deep learning and the decision-making ability of reinforcement learning [29]. Through RL, the network parameters of DL are tuned to continuously optimize their own behavior strategies; its framework is shown in Figure 2.

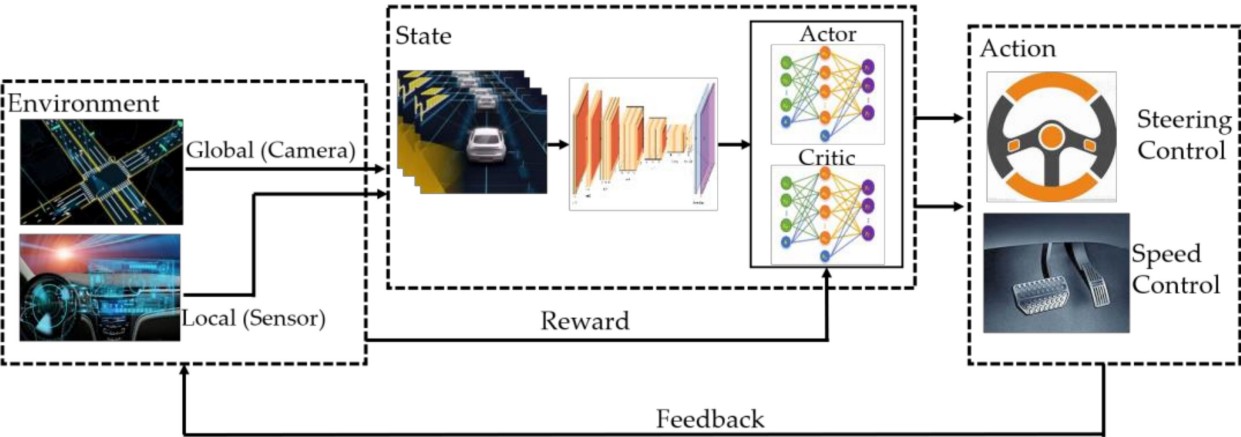

**Figure 2.** Deep reinforcement learning frame diagram.

This method has become a new research hotspot in the field of artificial intelligence and has been applied in fields such as robot control [30–32], autonomous driving [33], and machine vision [34–41]. In this paper, the vehicle autonomous driving decision problem is modeled with a partially observable Markov decision process (POMDP) [42], and the autonomous driving strategy optimization problem is solved by identifying the optimal driving strategy of the POMDP. Usually, the state $s_t$, action $a_t$, reward $r_t$, and next state $s_{t+1}$ are collected as a quadruple $s_t, a_t, r_t, s_{t+1}$ to form a set. Learning starts from time t until the end of time T and continuously optimizes its own strategy to maximize the cumulative reward $R_t$ Formula (1), where $\gamma \in [0,1]$ represents the discount factor, which is used to weigh the impact of future rewards on cumulative rewards.

$$R_t = \sum_{t'=t}^{T} \gamma^{t'-t} r_{t'} \tag{1}$$

### 2.3. Distributed Proximal Policy Optimization Algorithm

The DPPO algorithm [43] was proposed by Google's Deep Mind in 2017; its core idea is still the PPO algorithm. The PPO algorithm [44] is an RL algorithm that is based on the actor-critic framework proposed by Open AI. Its purpose is to solve the problem of slow updating of network parameters and the difficulty of determining the learning step size of the policy gradient algorithm (PG). Since the PG algorithm adopts the on-policy method to update the strategy, it needs to be resampled every time the network parameters are updated, resulting in a slow parameter update and difficulty in obtaining the optimal strategy. The off-policy method can learn from the samples generated by the old strategy every time the network parameters are updated, which ensures the comprehensiveness of the learning data and has stronger versatility and exploration. The PPO algorithm uses the importance sampling mechanism to transform the on-policy method into the off-policy method, realizes the reuse of the sampled data, and improves the update efficiency of the network Additionally, the advantage function is used to evaluate the advantage of the action value function in the sample trajectory and the value function of the current state. The advantage function $A_t$ is expressed by Formula (2), where $V(s_t)$ is the value function of the state $s_t$, and $\gamma \in [0,1]$ is the discount factor, which is used to weigh the future reward against the cumulative reward effects. If $A_t > 0$, the probability of taking the current action should be increased, and if $A_t < 0$, the probability of taking the current action should be

decreased. The objective function $J(\phi')$ of the critic network of the PPO algorithm is shown in Formula (3), where $\phi$ is critic network parameters.

$$A_t = \sum_{t'>t} \gamma^{t'-1} r_{t'} - V(s_t) \tag{2}$$

$$J(\phi) = -\sum_{t=1}^{T} \left( \sum_{t'>t} \gamma^{t'-1} r_{t'} - V_\phi(s_t) \right)^2 \tag{3}$$

The PPO algorithm uses the action probability ratio of the old and new strategies to $r_t(\theta)$ to represent the importance sampling weight, as shown in Formula (4), which limits the update range of the new strategy, reduces the sensitivity of the learning rate, and improves the stability of the algorithm. Its actor network objective function is shown in Formula (5).

$$r_t(\theta) = \frac{\pi_\theta(a_t|s_t)}{\pi_{\theta'}(a_t|s_t)} \tag{4}$$

$$J^{CPI}(\theta) = E_t \left[ \frac{\pi_\theta(a_t|s_t)}{\pi_{\theta'}(a_t|s_t)} A_t \right] = E_t(r_t(\theta)A_t) \tag{5}$$

where $a_t, s_t$ is the action and state of the t-th step; $\pi(*), V(*)$ is the strategy function and evaluation function; $\theta$, is the actor network parameters; $\pi_\theta, \pi_{\theta'}$ is the new strategy and old strategy; $E_t$ is the experience expectations for time step.

To constrain the update range of the old and new strategies of the objective function of the actor network, the PPO algorithm adds a clipping item and uses it to automatically clip the objective function, as shown in Formula (6), where the clip function constrains the importance sampling weight $r_t(\theta)$ to be in the range of $(1 - \varepsilon, 1 + \varepsilon)$. The min function obtains the minimum value of the original item and the truncated item, limits the update range of the ratio between the new policy and the old policy, and avoids the policy update too fast to fail to converge or too slow to converge. This paper adopts the second method.

$$J^{CLIP}(\theta) = E_t[min(r_t(\theta)A_t, clip(r_t(\theta), 1 - \varepsilon, 1 + \varepsilon)A_t)] \tag{6}$$

where $\varepsilon$ is the truncation coefficient; $clip(*)$ is the truncation function.

The DPPO algorithm is based on the PPO algorithm and adds multiple threads as local networks. The algorithm includes a global network and multiple local networks and adopts a centralized learning and decentralized execution learning method. The global network is responsible for updating the actor and critic parameters, and the local network is responsible for collecting sample data. Its network processing framework is shown in Figure 3. During the training process, multiple local networks independently and simultaneously interact with the environment according to the strategy of the global network, collect data information, calculate the policy gradient according to Formula (6), and transfer it to the shared gradient area for storage. When the shared gradient area stores a certain amount, the global network obtains gradient information from the shared gradient area for learning and updates the network parameters. Next, multiple local networks share the updated policy parameters of the global PPO network and concurrently collect data in their respective environments and continue to cycle the above process until reaching the maximum number of training steps to complete the training task. The specific performance is that multiple agents are added to the simulation environment for parallel training, and the agents can communicate with each other and provide feedback between strategies, which greatly improves the efficiency of training and solves problems such as slow convergence.

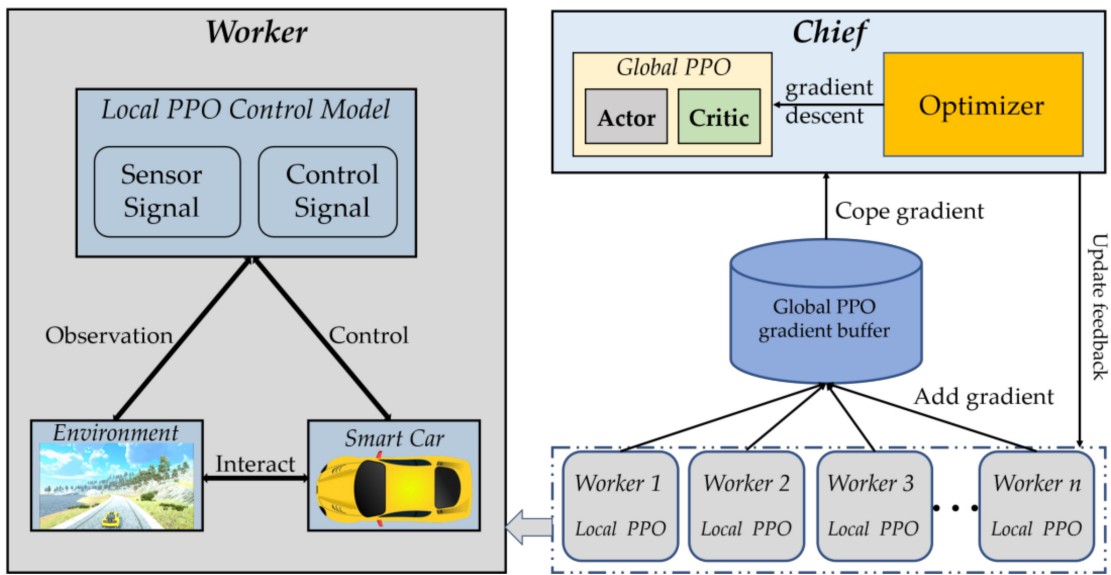

**Figure 3.** DPPO multithreaded processing framework.

## 3. Construction of an Automatic Driving Policy Learning System Based on APF-DPPO

Most of the current vehicle driving strategy learning methods use real vehicle road driving tests, which require high costs and high risks, are difficult to implement for special road conditions, and cannot effectively evaluate vehicle driving behavior strategies. Virtual reality and simulation technology provide a new test environment solution for autonomous driving testing, which can effectively reduce testing costs and risks. Therefore, this paper uses virtual reality technology to build a vehicle autonomous driving, decision-making simulation environment.

### 3.1. Introduction to the Simulation Environment

This paper adopts Unity ML-Agents as a reinforcement learning environment [45]. ML-Agents are an open-source plug-in for Unity. As shown in Figure 4, the overall architecture of Unity is mainly composed of a learning environment, an external communicator, and a Python API. The interactive communication between the learning environment and Python API is established through the Socket communication mechanism to achieve end-to-end decision control effects. The learning environment includes the Brain, Academy, and Agent. The agent in this paper is a virtual vehicle.

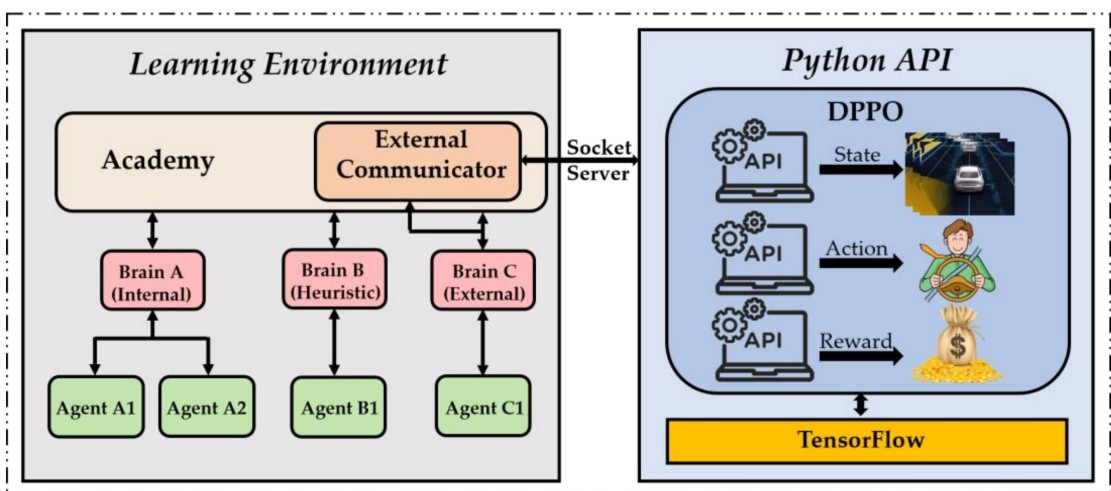

**Figure 4.** The overall architecture of the simulation environment.

### 3.2. State Space

During the training process, the state space needs to provide enough useful information to ensure effective learning of the virtual vehicle. The system obtains environmental information and vehicle status information through cameras and virtual ray sensors as the input signal of the network. According to the effective driving range of the virtual vehicle, the obstacles are limited to appearing randomly in the planned road area. To ensure a smooth training process, the collision of the positions of the virtual vehicle, obstacles, and obstacles on both sides of the lane should be avoided. In a complex environment, to achieve effective training of virtual vehicle autonomous driving decisions in a learning environment, it is necessary to observe virtual vehicle status information, road condition location information, and obstacle information. Combined with the known information that can be obtained in the actual driving process, including the spatial coordinates of the virtual vehicle, lane lines, and obstacles, the position observation variable set $O_P$ is ($P_{car}$, $P_{lane}$, $P_{obs}$). To guide the virtual vehicle to smoothly and quickly reach its destination, it is necessary to input the driving speed, driving time, and steering angle information and to establish the vehicle observation variable set $O_C$ as ($C_v$, $C_t$, $C_\theta$). It is also necessary to establish the distance observation variable set $O_D$ as ($D_{dis}$, $D_{goal}$) to ensure that the virtual vehicle can safely complete the automatic driving task. To accelerate the convergence speed of the driving policy model, all observed variables are normalized, and the state space set $S_{obs}$= {$P_{car}$, $P_{lane}$, $P_{obs}$, $C_v$, $C_t$, $C_\theta$, $D_{dis}$, $D_{goal}$}, as shown in Table 1.

**Table 1.** List of state space collections.

| Observed Variable | Symbol | Description |
|---|---|---|
| Position Variable | $P_{car}$ | Virtual vehicle spatial location |
|  | $P_{lane}$ | Lane line spatial location |
|  | $P_{obs}$ | Obstacle space location |
| Vehicle Variable | $C_v$ | Virtual vehicle driving speed |
|  | $C_t$ | Virtual vehicle driving time |
|  | $C_\theta$ | Virtual vehicle steering angle |
| Distance Variable | $D_{dis}$ | Distance between the obstacle and the virtual vehicle |
|  | $D_{goal}$ | Distance from vehicle to destination |

### 3.3. Action Space

The action space of the virtual vehicle is set according to its physical structure. According to the driving operation mode of the real vehicle, the speed control and steering control of the virtual vehicle are set. In order to improve the convergence speed of the algorithm, combined with the characteristics of the physical structure of the real vehicle, the virtual vehicle will select the driving strategy according to the environmental state and output the steering wheel angle, brake pedal opening, and throttle force information of the vehicle based on normalization, so set the virtual vehicle action space set $A_C$= {$a_{steer}$, $a_{brake}$, $a_{speed}$}, as shown in Table 2.

### 3.4. Optimizing the Reward Function Based on APF-DPPO

After initializing the state and action, the vehicle will randomly select the driving action strategy based on the environment information, but it cannot evaluate the merit of the action strategy based on the state variables. A reasonably designed reward function can effectively evaluate the behavioral decision of the vehicle and improve the learning efficiency and decision-making ability of the vehicle. However, in practical tasks, the sparse reward problem often exists, which tends to lead to slow algorithm iterations and even difficult convergence. The artificial potential field method is able to detect obstacles

and plan reasonable and stable, smooth, and continuous paths in complex unknown environments in real time to meet the requirements of autonomous driving strategies. To solve this problem, this paper establishes a continuous type reward function learning model based on the ideas of target attraction and obstacle repulsion of the artificial potential field method, and designs a destination guidance function, an obstacle avoidance function, and a time penalty function.

**Table 2.** List of action space sets.

| Way to Control | Description | Ranges |
|---|---|---|
| Speed Control | Accelerator pedal strength $a_{speed}$ ($a_{speed} > 0$ is acceleration, $a_{speed} = 0$ no operation) | [0,1] |
| | Brake pedal opening $a_{brake}$ ($a_{brake} > 0$ for braking, $a_{brake} = 0$ for no operation) | [0,1] |
| Steering Control | Steering wheel angle $a_{steer}$ ($-1 < a_{steer} < 0$ to turn left, $0 < a_{steer} < 1$ to turn right) | [−1,1] |

### 3.4.1. Destination Guide Function

In the traditional APF method, the position of the object determines the magnitude of the gravitational force, while in DRL, the reward signal determines the magnitude of the gravitational force, and its value is determined by the behavior of the agent. In this paper, a combination of the direction reward function $R_{dir}$ and distance reward function $R_{dis}$ is employed as the destination guidance function $R_{guide}$. When the driving state changes, the vehicle will move along the lane. If the speed direction of the vehicle is the same as the direction of the lane, the system will give a reward, and the reward value is inversely proportional to the steering angle of the vehicle; otherwise, the system will give a penalty. For example, in a certain state $S_t$, the speed of the virtual vehicle is $V_t$, the central axis of the road is $l$, and the angle formed by the direction of the vehicle speed and the central line of the road is the steering angle, which is represented by $\theta_t$, as shown in Figure 5. Figure 5 shows that the current state vehicle speed information is expressed by Equation (7), where $V_z$ is the longitudinal speed and $V_x$ is the lateral speed.

$$\begin{cases} V_z = V_t \cos \theta_t \\ V_x = V_t \sin \theta_t \end{cases} \tag{7}$$

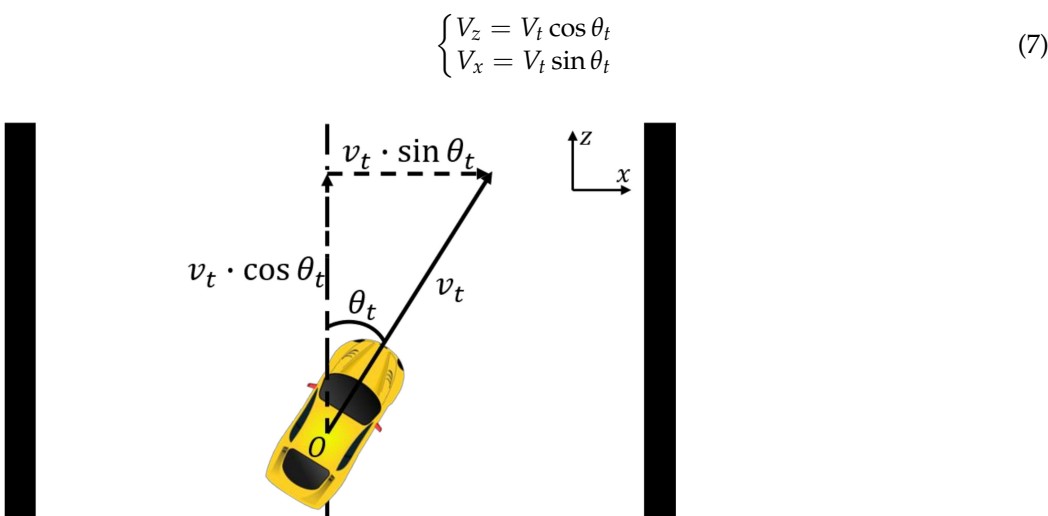

**Figure 5.** Schematic of the vehicle steering angle.

If the steering angle $\theta$ of the vehicle gradually decreases during the training process, the driving action strategy selected by the virtual vehicle in the current state is correct. At this time, the system will give a low reward according to the direction reward function; otherwise, the system will give a low penalty. When the steering angle $\theta$ is 0, the driving

strategy selected by the virtual vehicle is the best in the current state, and the system will give the highest reward and encourage the virtual vehicle to choose this direction strategy for driving. The direction reward function $R_{dir}$ is shown in Formula (8).

$$R_{dir} = \begin{cases} k_1 V_z & (\theta_{t+1} \neq 0 \cap \theta_t > \theta_{t+1}) \\ -k_2 V_x & (\theta_{t+1} \neq 0 \cap \theta_t < \theta_{t+1}) \\ k_3 & (\theta_{t+1} = 0) \end{cases} \tag{8}$$

$$R_{dis} = (\frac{D_{s_0}}{D_{s_0} + D_{s_t}})k_4 \tag{9}$$

$$R_{guide} = R_{dir} + R_{dis} \tag{10}$$

where $k_1$, $k_2$, $k_3$, $k_4$ are bootstrap reward function constants; $D_{s_0}$ is the distance required for the vehicle to reach the destination position in the initial state; $D_{s_t}$ is the distance required for the vehicle to reach the destination location in the current state.

As the virtual vehicle explores its direction, its position also changes so that the virtual vehicle gradually drives toward its destination. To determine whether the virtual vehicle has reached the destination, the distance reward function of the vehicle $R_{dis}$ is designed. When $D_{s_t}$ gradually shrinks, the vehicle is moving toward the destination, and a low reward is given according to Formula (9). When $D_{s_t}$ is 0, the vehicle has reached the destination, a high reward is given and the round ends. The destination guidance function is shown in Formula (10).

### 3.4.2. Obstacle Avoidance Function Setting

In the traditional APF method, when an object enters the action range of the potential field of the obstacle, it will be affected by the repulsive force. The obstacles in this paper are divided into two types: obstacles randomly generated in the road plane and obstacles on both sides of the lane that prevent the vehicle from driving out of the driving range. These obstacles will affect the vehicle's optimal driving decisions and reduce the smoothness and comfort of the vehicle's driving. When setting the collision avoidance function of virtual vehicle automatic driving strategy learning based on the idea of the APF method, the penalty signal determines the size of the repulsion force. As shown in Figure 6, in this case, the optimal driving strategy of the virtual vehicle should be the dashed driving trajectory, but its driving trajectory is deflected due to the influence of the repulsive force of the obstacle.

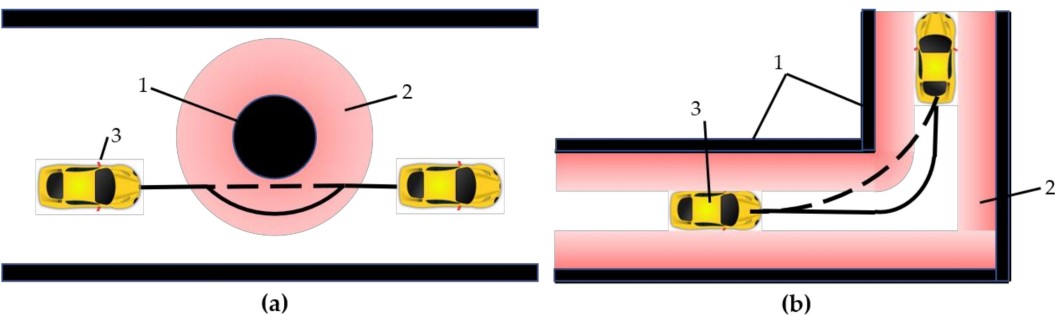

**Figure 6.** Schematic of the limitations of the traditional artificial potential field method. (**a**) is the range repulsion effect generated by randomly generated obstacles in the road, (**b**) is the range repulsion effect generated by the obstacles on both sides of the road that prevent vehicles from driving out. (**1**) is the obstacle, (**2**) is the range repulsion, and (**3**) is the virtual vehicle. The dotted line is the optimal path trajectory, and the solid line is the actual vehicle driving trajectory.

To solve the above problems, this paper regards the driving behavior of the virtual vehicle as X-Z plane motion. Movement along the Z axis means that the vehicle turns left and right, and movement along the X axis means that the vehicle turns forward and

backward. When the action of the virtual vehicle changes so that the virtual vehicle gradually approaches the obstacle, the system will give a penalty. This penalty value is negatively correlated with the approaching distance. As in state $t$, the distance between the virtual vehicle position $P_{car} = (x_t, z_t)$ and the obstacle position is $P_{obs} = (x_o, z_o)$, as denoted by $D_{r_i}$. During the training process, the ray sensor will calculate the distance information $D_{r_i}$ between the virtual vehicle and the obstacle in real time according to Formula (11), as shown in Figure 7. At this time, the system will obtain the minimum collision distance $D_{min}$ according to Formula (12) and determine whether the virtual vehicle may collide with the obstacle. If possible, the direction strategy will be punished according to Formula (13), and the round will be ended. Otherwise, no punishment will be made to avoid unnecessary punishment of the virtual vehicle during the automatic driving process, thereby improving the decision-making efficiency of autonomous collision avoidance driving of virtual vehicles.

$$D_{r_i} = \sqrt{(x_t - x_o)^2 + (z_t - z_o)^2} \tag{11}$$

$$D_{\min} = \min(D_{r_i}, D_{\min}) \tag{12}$$

$$R_{obs} = \begin{cases} -k_t e^{-D_{\min}} & (C \cap O_{obs} \neq \varnothing) \\ 0 & (C \cap O_{obs} = \varnothing) \end{cases} \tag{13}$$

where $k_t$ is the direction penalty coefficient; $C, O_{obs}$ is the virtual vehicle space collections and obstacle space collections.

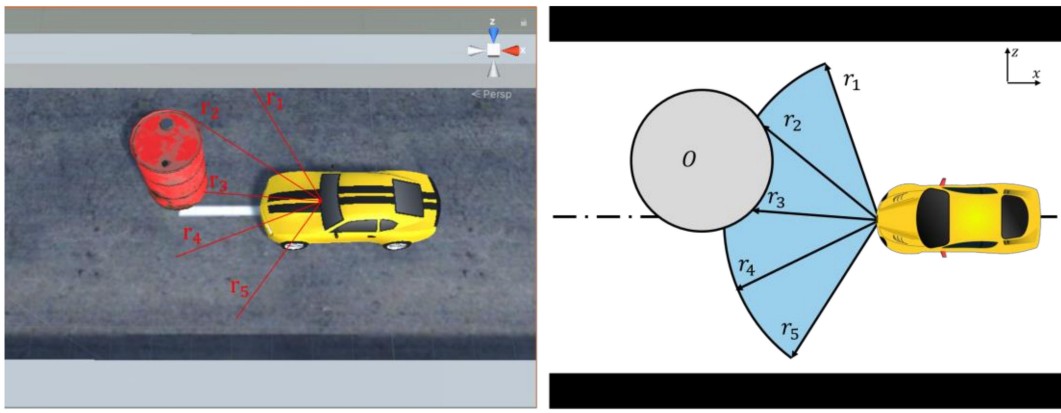

**Figure 7.** Using ray sensors to perceive learning environment information. The **left** is the actual vehicle encountering obstacles during driving in the virtual environment, and the **right** is the geometric schematic diagram of the left picture. The blue fan-shaped area is the sensing area of the sensor, and $(r_1, r_2, r_3, r_4, r_5)$ is the ray emitted by the sensor.

In a complex environment, the random and nonlinear distribution of obstacles renders the decision-making of vehicle driving behavior diverse and uncertain. To improve the efficiency of virtual vehicle driving decision-making, this paper uses the envelope method to simplify the motion model between the vehicle and the obstacle and transforms the collision problem into the intersection of plane geometric figures. By establishing the motion collision model of the virtual vehicle and by analyzing the collision results, it is further determined whether to give directional penalties to improve vehicle driving safety. First, it is necessary to determine whether the vehicle speed direction, obstacle position, and lane centerline are collinear.

1.  The vehicle speed direction, obstacle position, and road centerline are collinear, as shown in Figure 8a. A schematic of the driving trajectories of obstacles and virtual vehicles is shown in Figure 8b, where $O$ is the center point of the obstacle and $R_{obs}$ is the radius of the obstacle.

Figure 8b shows that the vehicle safety domain angle $\varphi$ can be obtained according to the trigonometric function relationship, as shown in Equation (14). When the steering angle $\theta$ of the vehicle is less than the safety domain angle $\varphi$, the vehicle has collided with the obstacle in the current state. At this time, the system will impose a maximum penalty on the decision-making direction and end the round. When the steering angle $\theta$ of the vehicle is greater than or equal to the safety domain angle $\varphi$ value, the vehicle has collided with the obstacle. The vehicle can autonomously avoid obstacles and continue driving, and the penalty function does not work at this time.

$$\varphi = \tan^{-1} \frac{R_{obs} + L_1}{D_t} \tag{14}$$

where $D_t$ is the distance from the center of the obstacle at the current moment to the center of the virtual vehicle; $L_1$ is the straight-line distance from the center of the virtual vehicle to the body surface along its horizontal axis.

2. The vehicle speed direction, obstacle position, and lane centerline are not collinear, as shown in Figure 9a. A geometric schematic of the intersection of the obstacle and the vehicle's path is shown in Figure 9b, where $O$ represents the current coordinate position of the obstacle and $R_{obs}$ is the radius of the obstacle.

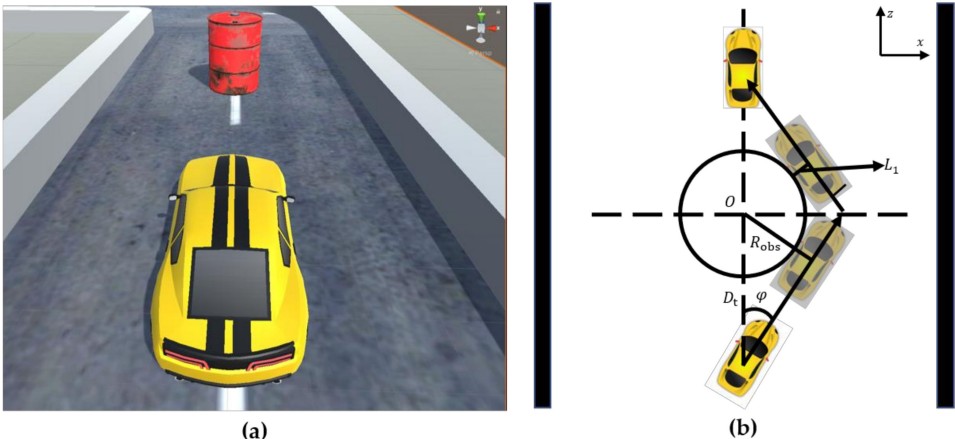

(a)                                                                                     (b)

**Figure 8.** When the direction of the vehicle, speed, the location of the obstacle and the center line of the lane are collinear (**a**). The obstacle is located at the central axis of the road during the actual driving of the vehicle (**b**), is the geometrical schematic diagram of the left figure.

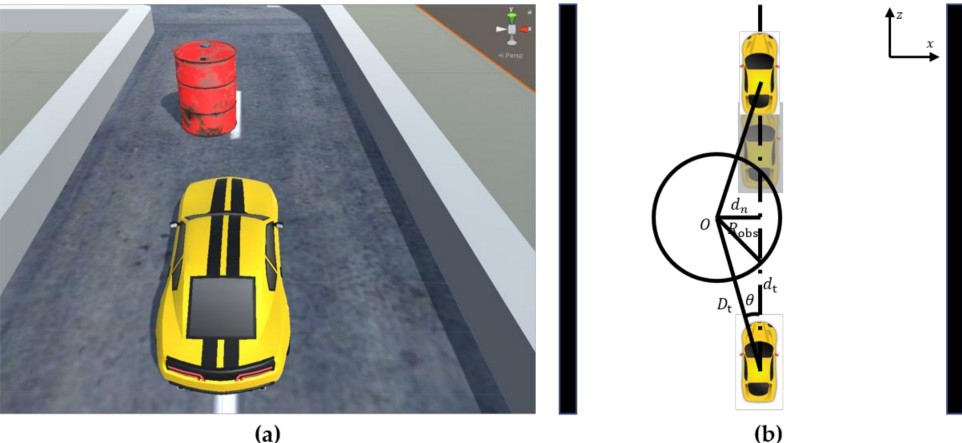

(a)                                                                                     (b)

**Figure 9.** When the direction of the vehicle speed, the position of the obstacle and the center line of the lane are not collinear. (**a**) is the actual vehicle when the obstacle is not on the central axis of the road, (**b**) is the geometrical schematic diagram of the left figure.

In the current state $S_t$, if the virtual vehicle collides with an obstacle, its precondition is described as shown in Formula (15), where $\kappa$ is the curvature of the obstacle and $d_n$ is the vertical projection line from the center point of the obstacle to the line segment $O_1O_2$.

$$\frac{d_n}{R_{obs}} \leq 1, R_{obs} = \frac{1}{\kappa}, d_n\kappa \leq 1 \tag{15}$$

Equation (16) can be obtained according to the law of cosines, where $\theta$ represents the steering angle of the virtual vehicle, $D_t$ is the distance from the center of the obstacle to the center of the virtual vehicle, and $d_t$ is the straight-line segment from the center of the vehicle to the intersection of the obstacle and the track segment:

$$R_{obs}^2 = D_t^2 + d_t^2 - 2Dd_t \cos\theta \tag{16}$$

If a collision occurs, that is, there will be an intersection between the radius circle of the obstacle and the driving path of the vehicle, there is at least one real solution to $d_t$. Substitute into Equation (15) and simplify to get:

$$d_t^2 - 2D_td_t \cos\theta + \left(D_t^2 - \left(\frac{1}{\kappa}\right)^2\right) = 0 \tag{17}$$

Simplify the root formula to get:

$$d_t = D_t \cos\theta \pm \sqrt{D_t^2(\cos^2\theta - 1) + \frac{1}{\kappa^2}} \tag{18}$$

$$D_t^2(\cos^2\theta - 1) + \frac{1}{\kappa^2} \geq 0, \frac{1}{\kappa} \geq d\sin\theta \tag{19}$$

$$d_n\kappa \leq 1 \tag{20}$$

The preconditions for collision are met. Therefore, in the current state $S_t$, the virtual vehicle will collide with the obstacle, and the system will punish the strategy direction and end the round.

### 3.4.3. Time Penalty Function Setting

To fully consider the economy of the vehicle driving strategy, a time penalty function is designed, and the time penalty coefficient $k_5$ is introduced to guide the virtual vehicle to reach the destination quickly, as shown in Equation (21).

$$R_{time} = -k_5C_t \tag{21}$$

where $k_5$ is the time penalty coefficient; $C_t$ is the time spent by the vehicle in the driving task.

In summary, the reward function $R$ of this system is shown in Formula (22), and the overall logic is shown in Figure 10.

$$R = R_{guide} + R_{obs} + R_{time} \tag{22}$$

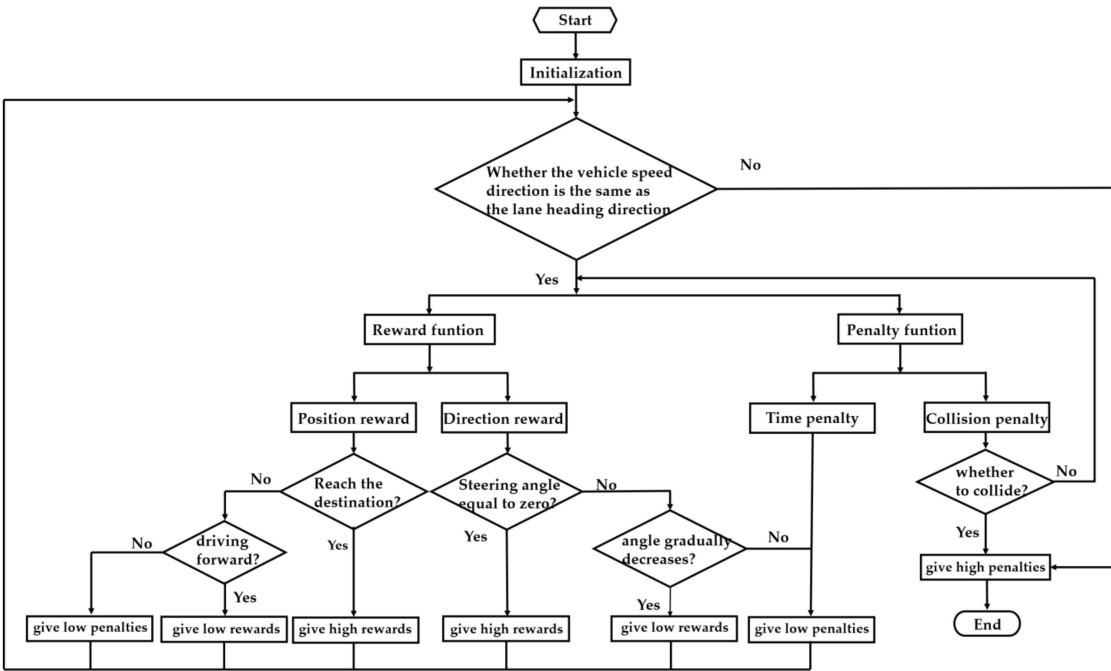

**Figure 10.** Logic diagram of reward function.

*3.5. APF-DPPO Algorithm Network Training Process*

The DPPO algorithm is a reinforcement learning algorithm based on the actor-critic (AC) architecture, including a policy network and value network. The algorithm optimizes the agent's behavior strategy by using the policy network to approximate the policy function and evaluates the agent's behavior by using the value network to approximate the value function. The steps of the APF-DPPO Algorithm 1 proposed in this paper are expressed as follows:

---

**Algorithm 1**. APF-DPPO algorithm for vehicle autonomous driving decision-making.

---

**Input:** observation information $obs_t$ = [ $P_{car}$, $P_{lane}$, $P_{obs}$, $C_v$, $C_\theta$, $C_t$, $D_{dis}$, $D_{goal}$]
Initialize $obs_t$
Initialize the store memory D to capacity N
Initialize the actor network parameters $\theta$, $\theta\prime$ and critic network parameters $\varnothing^0$
**For** episode = 1, M **do**
    Reset the environment and obtain the initial state $s_1$
    **For** epoch t = 1,T **do**
        Select $a_t = \{a_{steer}, a_{brake}, a_{speed}\}$ according to $\pi_{\theta\prime}(s_t)$
        Obtain reward $r_t = R_{guide} + R_{obs} + R_{time}$ and state $s_{t+1}$
        Collect $s_t, a_t, r_t$
        Store the transition ($s_t$, $a_t$, $r_t$, $s_{t+1}$) in D
        Update the state $s_t = s_{t+1}$
        Compute advantage estimates $A_t$ using generalized advantage estimation
        After every L steps
        Update $\varnothing$ by a gradient method using Equation (3)J($\varnothing$)
        Update $\theta$ by a gradient method using Equation (6)$J^{CLIP}(\theta)$
    **If** the vehicle hits an obstacle
    **Else if** the vehicle drives out of the boundary
    **Else if** the vehicle travels in the opposite direction
        Cancel this action
    **If** $T_i > T_{max}$ (where $T_{max}$ is the maximum step) **then**
        Finish training
    **End if**
        $\pi_{\theta\prime} \leftarrow \pi_\theta$
    **End for**
**End for**

---

## 4. Experiment and Evaluation

To verify the feasibility and effectiveness of the APF-DPPO proposed in this paper, a large number of experiments are designed. First, this section introduces the design of the autonomous driving policy learning scheme and analyzes the model training result data in detail. Second, the model results learned by the APF-DPPO method are verified by comparative experiments under different conditions, and the experimental results are analyzed. Last, the control effects of actual vehicle autonomous driving decision-making under different reward functions are objectively discussed.

### 4.1. Driving Strategy Learning Program Design

In this paper, the experimental platform configuration information is presented as follows: NVIDIA GTX1650 graphics card with 4G memory and AMD 4800H processor with a main frequency of 2.90 GHz; the simulation environment is Unity 2019.4.16f1c1. TensorFlow 2.0 and Cuda10.1 are selected to build the neural network computing environment. The training environment consists of straight, right, and left-turn road scenes. Simultaneously, 16 agents are set up for parallel training; orthogonal cameras and ray sensors are used to obtain training environment information; and a continuous action output space, randomized initial network parameters, and lightweight training environment layout are established, as shown in Figure 11.

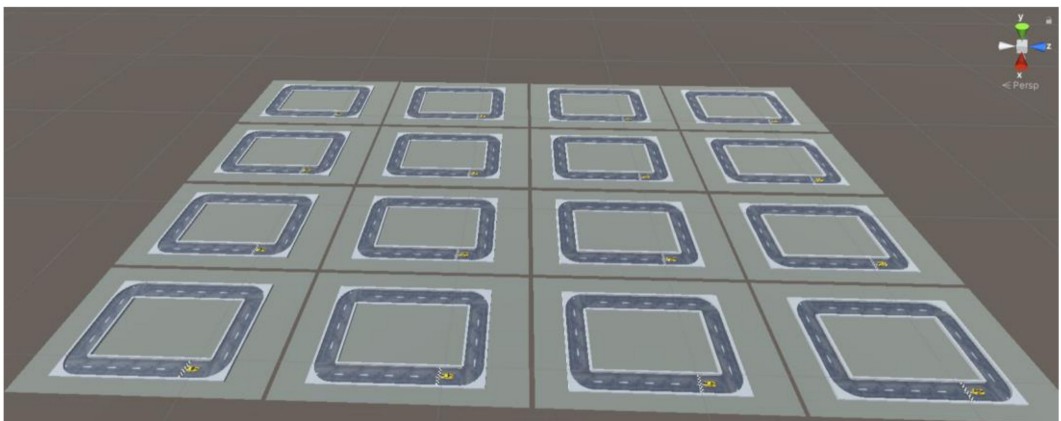

**Figure 11.** Vehicle autonomous driving policy training learning environment.

To ensure the smooth progress of training, the initialization state is set: the virtual vehicle is reset to the position of the nearest middle line from the lane and the random generation strategy of obstacles is applied; the termination conditions of the training round are set: the vehicle drives in reverse, collides with obstacles, or drives out of the lane, which is beneficial for reducing the invalid data generated in the experience pool and improving the training learning speed. The parameters of the training process are shown in Table 3.

**Table 3.** Training parameter settings.

| Parameter | Value |
| --- | --- |
| Batch Size | 1024 |
| Buffer Size | 10,240 |
| Learning Rate | $3.0 \times 10^{-4}$ |
| Beta β | $5.0 \times 10^{-3}$ |
| Epsilon ε | 0.2 |
| Lambda λ | 0.95 |
| Gamma γ | 0.99 |
| Num Epoch | 3 |
| Num Layers | 2 |
| Hidden Units | 1 28 |
| Max Steps | $5.0 \times 105$ |

## 4.2. Data Analysis of Training Results

Based on the designed experimental scheme, the virtual vehicle is trained on the autonomous driving strategy. The training process and results can be obtained through TensorBoard, and the experimental data can be downloaded and imported into MATLAB for drawing, as shown in Figure 12. Figure 12a shows that with an increase in the number of training steps, the cumulative reward value obtained by the virtual vehicle also gradually increases, and the reward value rises the fastest at 300,000 to 450,000 steps, indicating that the virtual vehicle can quickly choose the correct driving decision and reach the convergence state when it is near 480,000 steps. At this time, the vehicle can quickly, safely, and stably drive along the lane, indicating that it can obtain a better driving decision control effect. The entropy value of the vehicle driving strategy in Figure 12b shows a sharp upward trend at the beginning of training. With an increase in the number of training steps, its value reaches the maximum value and then decreases slowly, indicating that the virtual vehicle after learning and training has the ability to autonomously make decisions. As the training time increases, the maximum steps for the training algorithm to search for the optimal strategy gradually decrease, as shown in Figure 12c, which shows that the virtual vehicle can accurately select the correct driving strategy in a short time. In Figure 12d, the policy loss of the virtual vehicle also trended down during correct training, further demonstrating the effectiveness of the method.

## 4.3. Model Validation and Result Analysis Based on Transfer Learning

To further verify the effectiveness and advancement of the APF-DPPO algorithm, combined with the transfer learning method, the training result model was transferred to the comparative experiment for driving decision inference learning and result testing. Considering the influence of lane type, obstacle type, and reward function on the performance of the autonomous driving policy model, three sets of comparative experiments are set up: a generalization performance experiment of the vehicle automatic driving policy model in different lane environments, an experiment of the influence of different types of obstacles on vehicle automatic driving decision-making, and a comparison experiment of vehicle automatic driving decision-making performance under different reward functions.

### 4.3.1. Generalization Performance Experiment of the Autonomous Driving Model in Different Lane Environments

To verify the generalization performance of the autonomous driving policy model learned based on the APF-DPPO algorithm, different lane environments are designed to conduct vehicle automatic driving decision inference learning experiments. The verification lanes are randomly combined with L-shaped, U-shaped, and S-shaped curves. According to the complexity of the lane trajectory, the verification lanes can be divided into six categories, as shown in Figure 13. Among them, the easy lane is the learned lane for the training in this paper.

The training result models are transferred to different types of environments for virtual vehicle autonomous driving decision inference experiments. The simulation results are shown in Table 4. The length of the lane refers to the actual distance of the lane; the driving time refers to the time required to complete the five-lap driving task; the average reward value refers to the ratio of the accumulated reward value obtained to the number of laps of the driving task; and the pass rate refers to the ratio of the number of completed driving tasks to the total number of tasks. As the complexity of the lane environment deepens, the lane length increases, and the turning road conditions or randomly generated obstacle distances form a narrow space with both sides of the road in the subsequent automatic driving process continuously, resulting in the virtual vehicle wandering back and forth in that road condition and consuming more time. At the same time, it can be seen that the virtual vehicle gets the average reward value and the lane length are positively correlated, which indicates that the method in this paper has better stability and robustness. From the passing rate, it can be seen that the virtual vehicle can complete the automatic driving

task with 100% passing rate in different types of lane environments, which has good generalization ability.

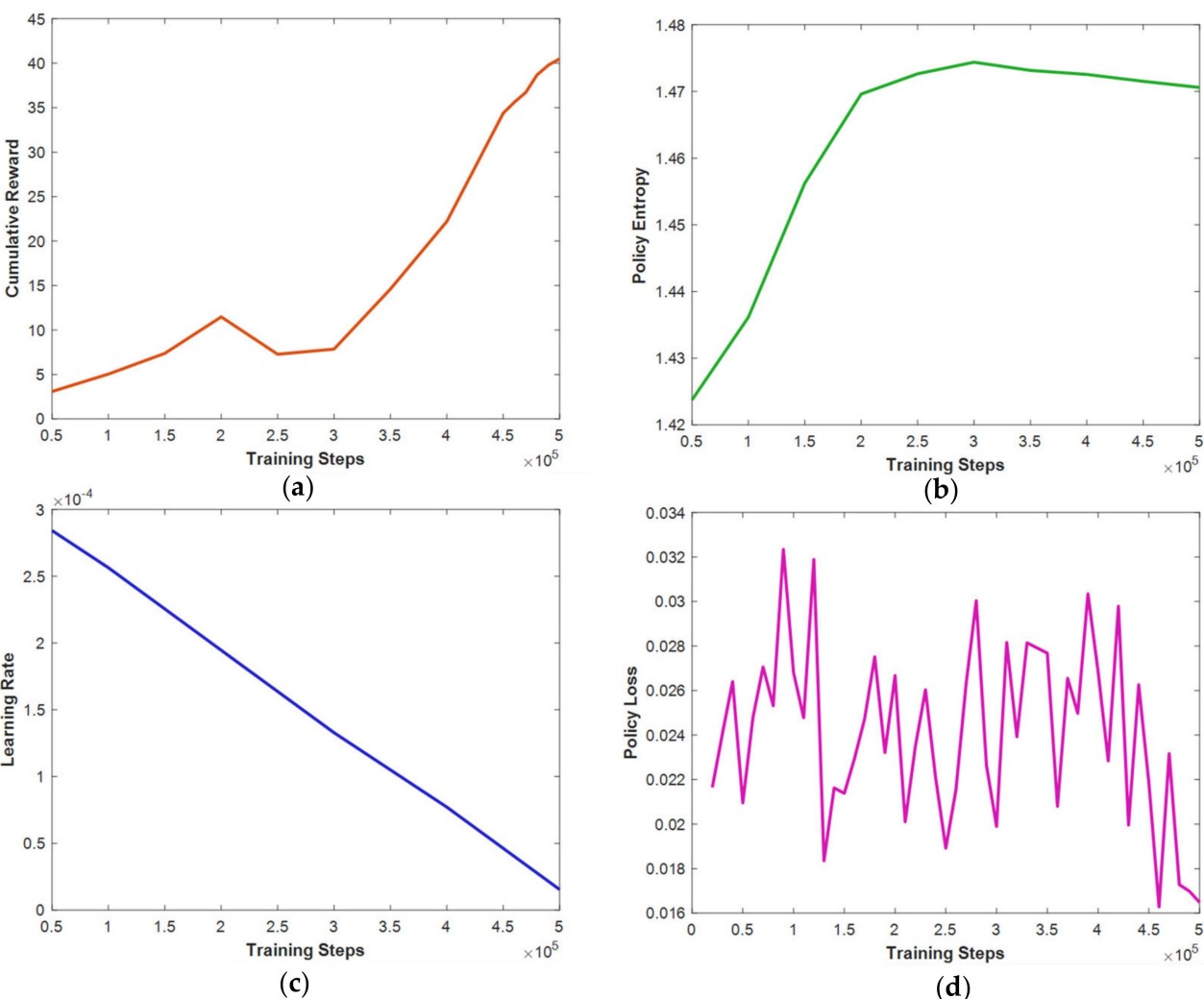

**Figure 12.** Data results for deep reinforcement learning training learning. (**a**) is the trend graph of the cumulative reward value, (**b**) is the trend graph of the entropy value of the strategy, (**c**) is the trend graph of the step length required to explore the optimal strategy, (**d**) is the trend graph of the average loss of the strategy update.

4.3.2. Influence of Different Types of Obstacles on Vehicle Autonomous Driving Decisions

To further verify the robustness of the autonomous driving policy model trained based on the APF-DPPO algorithm, different types of obstacles are designed to conduct vehicle autonomous driving decision inference learning experiments. According to the random generation strategy of obstacles, two different driving scenarios are established, as shown in Figure 14. The red object in Figure 14a is the obstacle that generates the penalty feedback, that is, once the virtual vehicle enters the obstacle warning area during the driving process, the obstacle collision avoidance penalty function will give feedback according to the degree of approach. In Figure 14b, the green objects are obstacles that do not generate penalty feedback, that is, the virtual vehicle enters the obstacle warning area during driving, and the obstacle collision avoidance function does not give any penalty feedback.

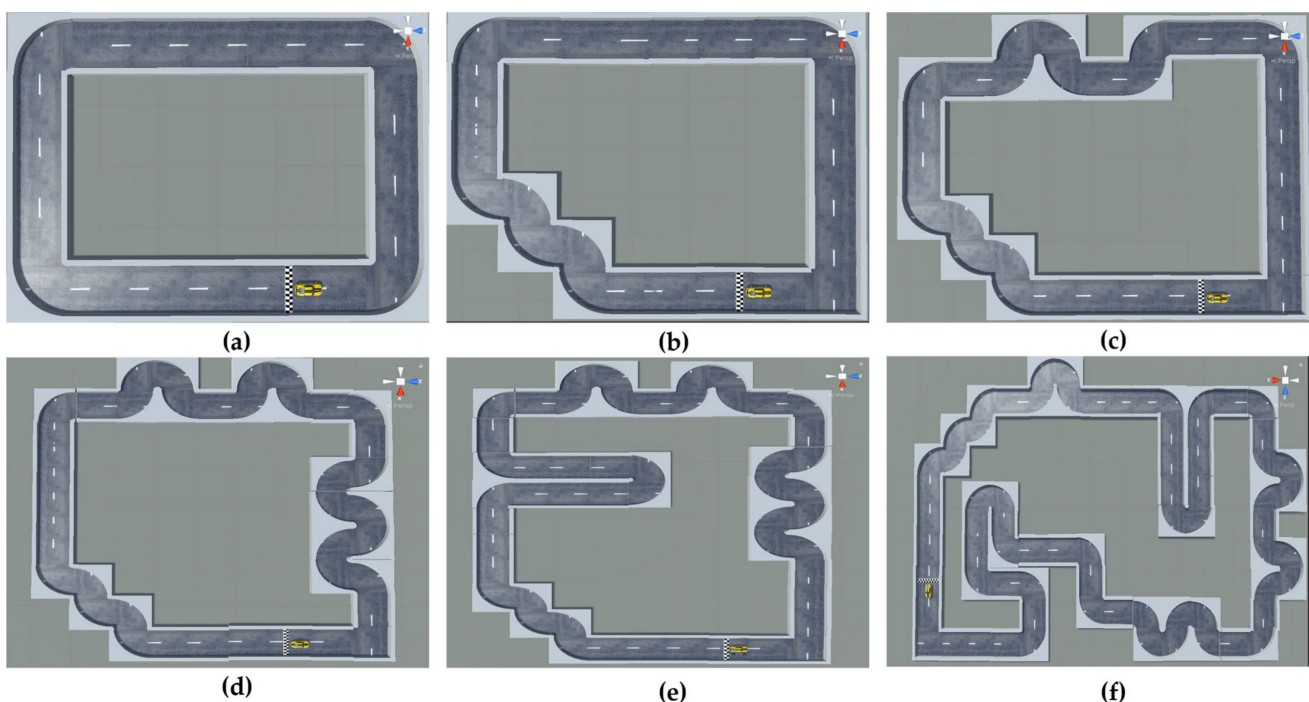

**Figure 13.** Different types of verification lanes. (**a**) is an easy lane type, (**b**) is a relatively easy lane type, (**c**) is a common lane type, (**d**) is a difficult lane type, (**e**) is a relatively difficult lane type, and (**f**) is a particularly difficult lane type.

**Table 4.** Experiment on generalization performance of virtual vehicle autonomous driving under different road conditions.

| Lane Type | Lane Length/cm | Driving Time/s | Average Reward Value | Pass Rate/% |
|---|---|---|---|---|
| Easy lane | 20 | 131 | 19.6 | 100 |
| Easier lane | 25 | 148 | 24.2 | 100 |
| General lane | 30 | 189 | 28.2 | 100 |
| Difficult lane | 40 | 236 | 37.8 | 100 |
| More difficult lane | 50 | 313 | 47.2 | 100 |
| Particularly difficult lanes | 70 | 445 | 66.8 | 100 |

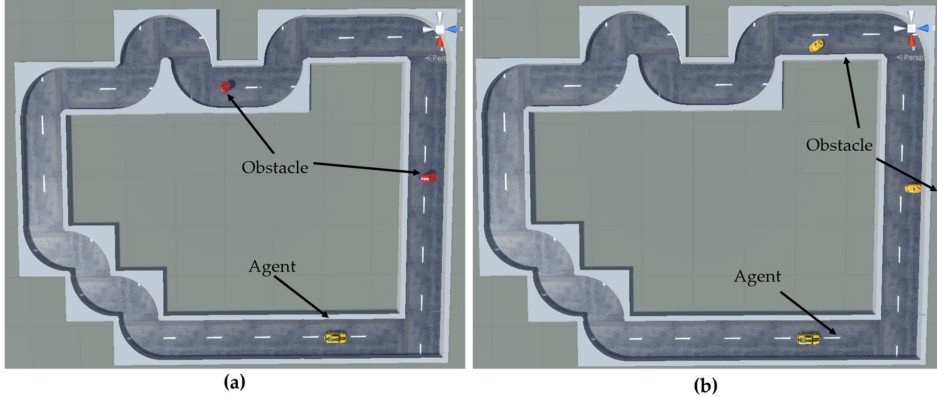

**Figure 14.** Autonomous driving obstacle avoidance experiments of virtual vehicles in different types of obstacle environments. (**a**) is the obstacle environment with penalty feedback, (**b**) is the obstacle environment without penalty feedback.

The training result models were transferred to different obstacle environments for virtual vehicle autonomous driving decision inference experiments. The simulation results are shown in Table 5. The maximum completion rate of the virtual vehicle in the obstacle

environment with punishment feedback is 96.3%, which is 3.8% higher than that in the environment without punishment feedback. The system optimizes its own driving strategy and has the ability to autonomously avoid obstacles. The results show that the system can complete the obstacle avoidance driving task in most cases and meet the needs of automatic driving technology. Among them, three driving tasks failed because the randomly generated obstacles were located at the turn of the lane and formed a narrow space with the obstacles on both sides of the road. The virtual vehicle fell into the narrow space during the forward process. At this time, the obstacles generated collision avoidance penalty signals and time penalties. The sum of the collision avoidance penalty signal generated by the obstacle and the time penalty signal may be equal to the reward signal generated by the destination guidance function, causing the virtual vehicle to get stuck in this position and be unable to continue driving.

**Table 5.** Different types of obstacles affecting the vehicle automatic driving performance experiment.

| Obstacle Type | Vehicle Driving Situation | | Completion Rate/% |
|---|---|---|---|
| | Number of Completions | Number of Failures | |
| Generate punitive feedback | 77 | 3 | 96.3 |
| No punitive feedback | 74 | 6 | 92.5 |

4.3.3. Experiments on the Decision-Making Performance of Autonomous Driving with Different Reward Functions

To verify the effectiveness of the method in this paper compared with other methods, different reward functions are designed to compare the learning performance of virtual vehicle autonomous driving strategies. There are four types of reward functions: ① Reward function without collision penalty term, that is, when the virtual vehicle collides with an obstacle during the training process, the penalty function does not work. ② The reward function for the contact collision penalty, that is, the penalty function, only works when the virtual vehicle is in contact with the obstacle during the training process. ③ Reward function based on the APF method, that is, the penalty function works when the virtual vehicle enters the obstacle warning area during the training process, and the penalty value is inversely proportional to the distance from the virtual vehicle to the obstacle. ④ Improved reward function of APF method, that is, the method in this paper. The destination steering function is the same for all reward functions. These four reward functions are employed for 500,000-step driving policy training and learning; the training results are shown in Figure 15a. Figure 15a shows that the reward function method without the collision penalty item obtains the lowest cumulative reward value, and the virtual vehicle spends a considerable amount of time in the unnecessary direction exploration process during the training and learning process. Compared with other methods, the accumulated reward value obtained by designing the reward function based on the APF-DPPO algorithm is the highest, indicating that the virtual vehicle can select the optimal driving strategy in a short time.

To further verify the stability of the method in this paper compared with other methods, the latest training result model is transferred to the general lane environ-ment for the automatic driving strategy inference learning experiment and result test. Within a fixed driving time of 500 s, the variation curve of the cumulative reward value obtained by the driving decision model of the virtual vehicle under different reward functions is shown in Figure 15b. The results show that the method in this paper is relatively optimal in vehicle autonomous driving decision-making and driving stabil-ity. Due to the complexity and randomness of the road scene, the driving strategy se-lected each time is variable. The contact collision penalty mechanism method obtains a higher reward value from 0 to 200 s. However, the continuous collision between the virtual vehicle and the obstacle occurs at the subsequent continuous turns. At this time, the virtual vehicle is constantly exploring the best driving strategy and wander-ing in place. The reward value obtained by the reward

function method based on the APF method is relatively low, which is mainly affected by the action range of the ob-stacle repulsion, resulting in a relatively longer driving time. The method in this paper can effectively reduce the occurrence of this phenomenon, and the cumulative reward value obtained within 500 s is the highest, which is 69 points higher than the reward function method based on the APF method. Its autonomous driving decision-making efficiency fluctuates only slightly under different road conditions. This finding shows that the method in this paper can effectively improve the decision-making speed of the automatic driving of vehicles.

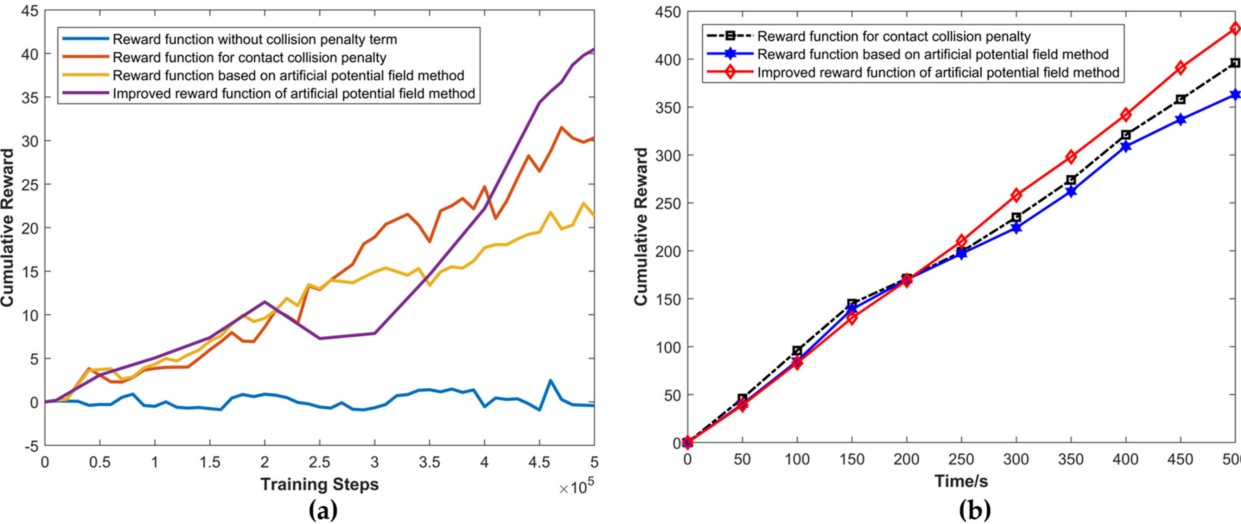

**Figure 15.** Variation trends of virtual vehicle training data and cumulative reward values under different reward functions. (**a**) is the training result of the virtual vehicle under different reward functions, (**b**) is the trend graph of the cumulative reward value obtained by the virtual vehicle under different reward functions in a fixed time.

*4.4. Decision Control Effect of Actual Vehicle Autonomous Driving*

Under different road conditions, different reward functions have different effects on the actual driving path of the vehicle's automatic driving decision, as shown in Figure 16. The red route trajectory is the automatic driving trajectory effect of the vehicle based on the reward function of the APF method, and the green route trajectory is the actual driving trajectory effect of the vehicle using the method in this paper. According to the figure, with the method in this paper, the vehicle automatic driving decision-making achieves a better decision-making control effect; the driving trajectory of the vehicle is improved; and the driving distance and driving time is greatly shortened.

For different types of obstacle environments, the actual driving path effect diagram of the vehicle's automatic driving decision is shown in Figure 17a,b. Red objects are obstacles that generate penalty feedback, and green objects are obstacles that do not generate penalty feedback. The figure shows that the virtual vehicle has higher driving smoothness and better comfort in the obstacle environment that generates punishment feedback. Figure 17c,d shows the effect of obstacles on the trajectory of the virtual vehicle's autonomous driving strategy under different positions. As shown in the figure, no matter where the obstacle is, the virtual vehicle can autonomously avoid the obstacle and successfully complete the driving task, which meets the needs of the vehicle's automatic driving technology. Figure 17e,f shows the effect of the actual driving path of an intelligent vehicle making automated driving decisions in an environment with conflicting vehicles. As can be seen from the figure, the intelligent vehicle (yellow car) is not only able to automatically avoid competing vehicles, but also obtains the best driving strategy, which significantly reduces the time to complete the driving task.

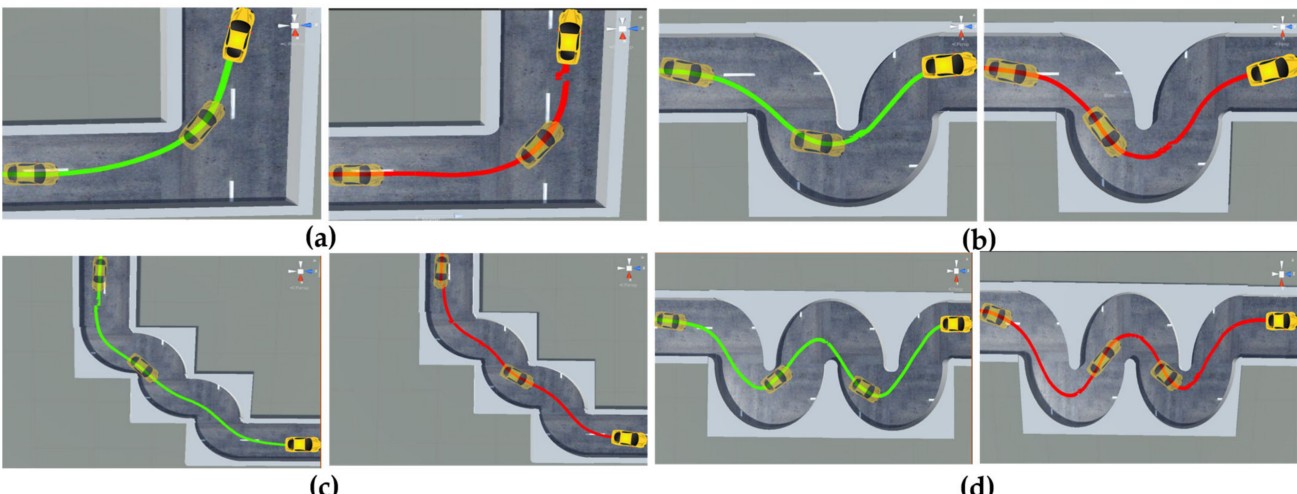

**Figure 16.** The actual effect of virtual car autonomous driving in different road conditions. (**a**) is an "L"-shaped road condition, (**b**) is a "U"-shaped road condition, (**c**) is an "S"-shaped road condition, and (**d**) is a "W"-shaped road condition.

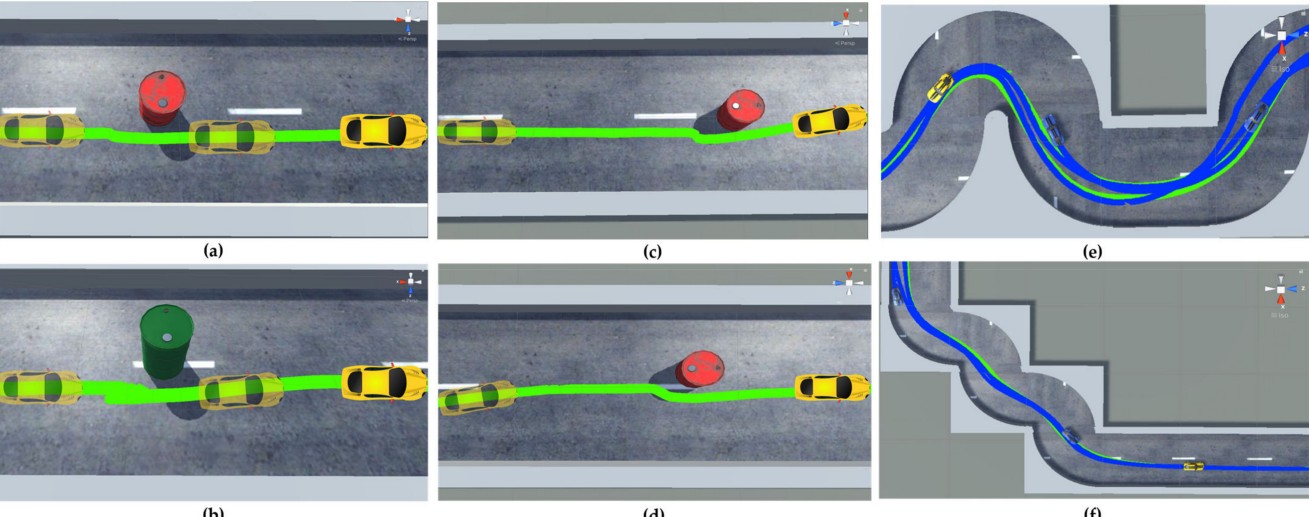

**Figure 17.** The actual effect of virtual vehicle autonomous driving in different types of obstacle environments. (**a**,**b**) are the effects of different types of obstacles on the autonomous driving strategy of virtual vehicles, (**c**,**d**) are the effects of obstacles in different positions that generate penalty feedback on the autonomous driving strategies of virtual vehicles, (**e**,**f**) are the actual driving process of two virtual vehicles with conflicting autonomy policies in the autonomous driving process.

To facilitate the viewing of the simulation process effect and test result data, a simulation running environment and an interactive user interface are built. The overall effect is shown in Figure 18. The system has basic functions, such as reset, run, and pause. Users cannot only select or define lane types but also set driving tasks, the type of vehicle with or without conflict, and the number of randomly generated obstacles for purposeful automated driving decision-making simulation experiments. Users can obtain relevant driving data information through the system interface, which can serve as a reference for actual vehicle autonomous driving decisions.

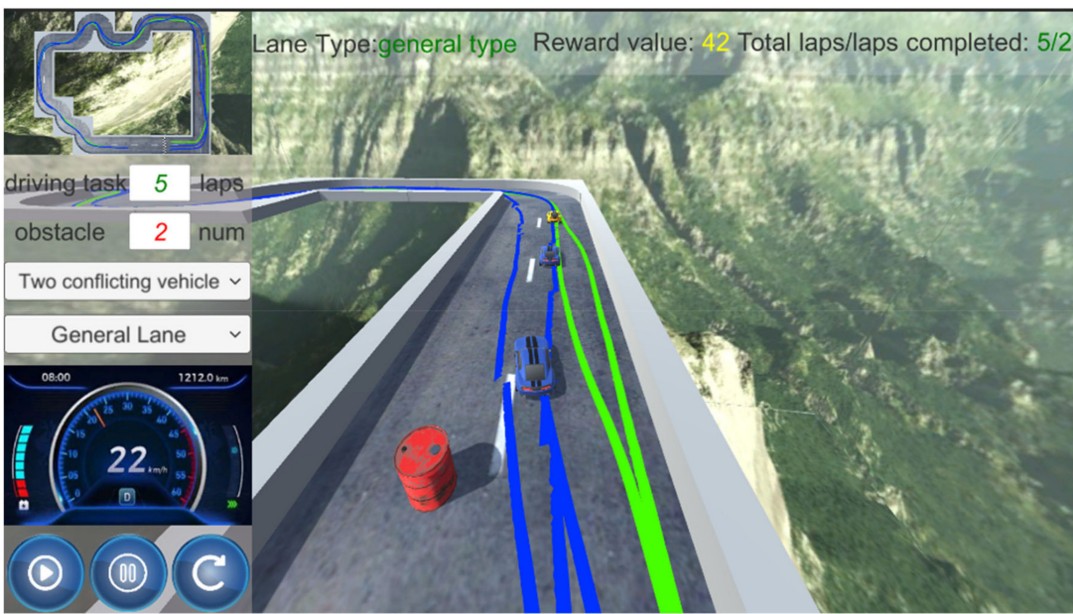

**Figure 18.** Overall effect of the system.

## 5. Conclusions

In this paper, an end-to-end autonomous driving policy learning method based on deep reinforcement learning is proposed to solve the behavioral decision problem of vehicle autonomous driving in complex and variable environments. The APF-DPPO learning model is established to optimize the vehicle self-driving policy learning. For the problem that the artificial potential field method has range repulsion to affect the optimal driving strategy, a directional penalty function method combining collision penalty and yaw penalty is proposed, and collision detection is performed for virtual vehicle driving behavioral actions, and the collision results are analyzed to selectively give penalties. A large number of comparison experiments are set up to verify the feasibility and effectiveness of the APF-DPPO learning model. The training result model is migrated to the control experiments using the migration learning method for automatic driving decision performance testing. The simulation experimental results show that the virtual vehicle can complete the full driving task with a 100% passing rate under different driving environments in the five-lap driving task; the completion rate of the virtual vehicle's automatic driving task under different types of obstacle environments is up to more than 92.5%. Under different reward and punishment mechanisms, the method in this paper obtains the highest cumulative reward value within 500 s, which improves 69 points compared with the reward and punishment mechanism method based on the artificial potential field method, and has higher adaptability and robustness under complex road conditions. The results show that compared with other methods, the driving strategy stability and decision control effect of this paper is better, but there is also a small probability of driving task failure in some special driving scenarios, which has some limitations. In future research, the curiosity mechanism will be introduced to improve the model training learning efficiency. The reward function mechanism of the virtual vehicle is further optimized for some demanding road conditions environments (such as continuous turns, etc.) and different shapes of obstacles and their states to improve the stability and robustness of the system.

**Author Contributions:** Conceptualization and methodology, H.W. and J.L.; validation, J.L., P.Z. and C.L.; data curation, Y.Z.; writing—original draft preparation, J.L., P.Z. and C.L.; writing—review and editing, H.W. and X.Z.; visualization, J.L.; supervision, H.W. and X.Z.; funding acquisition, H.W. All authors have read and agreed to the published version of the manuscript.

**Funding:** This work was supported in part by the Laboratory of Lingnan Modern Agriculture Project under Grant NT2021009 and the No. 03 Special Project and the 5G Project of Jiangxi Province under Grant 20212ABC03A27.

**Institutional Review Board Statement:** Not applicable.

**Informed Consent Statement:** Not applicable.

**Data Availability Statement:** Not applicable.

**Acknowledgments:** We sincerely acknowledge the project funding support. We are also grateful for the efforts of all our colleagues.

**Conflicts of Interest:** The authors declare no conflict of interest.

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
