# Peer review of "APF-DPPO: An Automatic Driving Policy Learning Method Based on the Artificial Potential Field Method to Optimize the Reward Function"

_machines, doi:10.3390/machines10070533_

Round 1
Reviewer 1 Report
The article proposes an end-to-end autonomous driving strategy learning method based on deep reinforcement learning, which can improve the autonomous decision-making efficiency of the vehicle.
The article is very interesting and well structured.
However, before publication, it requires a major update:
1) Improve the state of the art. For instance see ( [G. N. Bifulco, A. Coppola, S. G. Loizou, A. Petrillo and S. Santini, "Combined Energy-oriented Path Following and Collision Avoidance approach for Autonomous Electric Vehicles via Nonlinear Model Predictive Control," 2021 IEEE International Conference on Environment and Electrical Engineering and 2021 IEEE Industrial and Commercial Power Systems Europe (EEEIC / I&CPS Europe), 2021, pp. 1-6], [Gao, K., Yan, D., Yang, F., Xie, J., Liu, L., Du, R., & Xiong, N. (2019). Conditional artificial potential field-based autonomous vehicle safety control with interference of lane changing in mixed traffic scenario. Sensors, 19(19), 4199.], [Z. Zhu and H. Zhao, "A Survey of Deep RL and IL for Autonomous Driving Policy Learning," in IEEE Transactions on Intelligent Transportation Systems, doi: 10.1109/TITS.2021.3134702.])
2) Tables are extremely large. Please, improve their quality.
3) It would be interesting to add another example test scenario with two (conflicting) vehicles
4) A comparison with other approaches should be added.
Reviewer 2 Report
Dear Authors:
Thanks for your efforts.
The paper proposes an end-to-end autonomous driving strategy learning method based on deep reinforcement learning, which can improve the autonomous decision-making efficiency of the vehicle.
Here are some questions listed as follows:
Q1: In Abstract and Conclusion, What are the numerical improvements based on the experiments?
Q2: Throughout context, there are some English words and senstances are not properly used, such as "rule based" not "rule base", etc.
Q3: In Equations (2, 3), what are the definitions of V(.) and Phi'?
Q4: In Equation (5), what is the definition of Et?
Q5: In Tbale 2, the ranges of speed control and steering angle are not meaningful. The ranges are based on normalization? Please specify.
Q6: In Page 10, the paragraph just above 3.4.2, "0the" is not meaningful. Please rectify.
Q7: Please review this Algorithm 1 in terms of format and English.
Q8: What are the numerical improvements of the approach proposed?
Q9: What are the limitations and drawbacks of this approach?
Thanks and Best Regards
Reviewer 3 Report
Paper is important for autonomous vehicle concept. It needs revisions before publication-
(1) Author should write paper in your own words.
(2) What are motivations behind this research work?
(3) All tables and figures should be explained clearly.
(4) The English and typo errors of the paper should be checked in the presence of native English speaker.
(5) Author should add future scope of the paper.
(6) Why author choose Artificial Potential Field Method to Optimize the Reward Function? Author must explain.
(7) What are the important parameters which were chosen in Artificial Potential Field Method to Optimize the Reward Function? Author must explain.
(8) How simulation is achieved in the proposed research work? Author must explain in the paper.
(9) More explanation on figures and tables is expected.
(10) The abstract, conclusion and future scope should be clearly understood to the audience. Author must revise in the context of novelty and important findings of the proposed research work.
(11) The suggested papers must be cited in the paper, these are based on different applications and optimization research work-
(a) Actual deviation correction based on weight improvement for 10-unit Dolph–Chebyshev array antennas
(b) H–V scan and diagonal trajectory: accurate and low power localization algorithms in WSNs
(c) Integration of renewable energy sources, energy storage systems, and electrical vehicles with smart power distribution networks
(d) Elements Failure Detection and Radiation Pattern Correction for Time-Modulated Linear Antenna Arrays Using Particle Swarm Optimization
(e) A Non-Uniform Circular Antenna Array Failure Correction Using Firefly Algorithm
(f) ECG signal analysis using CWT, spectrogram and autoregressive technique
(g) Autonomous computation offloading and auto-scaling the in the mobile fog computing: a deep reinforcement learning-based approach
(h) Neuropathic complications: Type II diabetes mellitus and other risky parameters using machine learning algorithms
(i) Investigation on automated surveillance monitoring for human identification and recognition using face and iris biometric
(j) A reinforced random forest model for enhanced crop yield prediction by integrating agrarian parameters
(k) RL based hyper-parameters optimization algorithm (ROA) for convolutional neural network
(l) Prediction of atherosclerosis pathology in retinal fundal images with machine learning approaches
Round 2
Reviewer 1 Report
None
Reviewer 2 Report
Dear Authors:
Current version is fine.
Thanks and Best Regards
Reviewer 3 Report
Authors have done all recommended corrections. Now paper is acceptable in current form.